# Cj0683 Is a Competence Protein Essential for Efficient Initialization of DNA Uptake in *Campylobacter jejuni*

**DOI:** 10.3390/biom13030514

**Published:** 2023-03-11

**Authors:** Julia C. Golz, Sandra Preuß, Christoph Püning, Greta Gölz, Kerstin Stingl

**Affiliations:** 1National Reference Laboratory for Campylobacter, Department of Biological Safety, German Federal Institute for Risk Assessment, Diedersdorfer Weg 1, 12277 Berlin, Germany; 2Institute of Food Safety and Food Hygiene, Freie Universität Berlin, Königsweg 69, 14163 Berlin, Germany

**Keywords:** competence, genetic diversity, adaptation, natural transformation, *pilQ*, Cj1474c, Cj0011c, *comEC*, Cj1211, *Helicobacter pylori*

## Abstract

*C. jejuni* is an important food-borne pathogen displaying high genetic diversity, substantially based on natural transformation. The mechanism of DNA uptake from the environment depends on a type II secretion/type IV pilus system, whose components are partially known. Here, we quantified DNA uptake in *C. jejuni* at the single cell level and observed median transport capacities of approximately 30 kb per uptake location. The process appeared to be limited by the initialization of DNA uptake, was finite, and, finalized within 30 min of contact to DNA. Mutants lacking either the outer membrane pore PilQ or the inner membrane channel ComEC were deficient in natural transformation. The periplasmic DNA binding protein ComE was negligible for DNA uptake, which is in contrast to its proposed function. Intriguingly, a mutant lacking the unique periplasmic protein Cj0683 displayed rare but fully functional DNA uptake events. We conclude that Cj0683 was essential for the efficient initialization of DNA uptake, consistent with the putative function as a competence pilus protein. Unravelling features important in natural transformation might lead to target identification, reducing the adaptive potential of pathogens.

## 1. Introduction

*Campylobacter jejuni* is the leading cause of bacterial food-borne enteritis with more than 220,000 cases reported in 2019 in the European Union [1]. Human campylobacteriosis is characterized by acute diarrhea, abdominal cramps, fever, and nausea. However, long-term sequelae, such as the Guillain-Barré syndrome, can also occur in few cases [2,3]. *C. jejuni* displays a high genetic diversity due to its natural transformation capacity. This leads to an increase of the bacterium’s adaptive potential resulting in successful survival including the acquisition of antimicrobial resistances [4,5,6,7]. A prerequisite of natural transformation is the capacity to take up DNA from the environment. Competence development in *C. jejuni* occurred at neutral pH under microaerobic conditions and was shut-down at slight acidity as well as under atmospheric oxygen levels [8]. This suggested that natural transformation in *C. jejuni* was most active in its natural habitat, the intestinal tract of warm-blooded animals. 

DNA uptake in bacteria is a two-step process and the proteins implicated have been concisely reviewed [9,10]. In *C. jejuni*, the process of DNA uptake is mediated by homologs of a type II secretion/type IV pilus system [11,12]. *C. jejuni* harbors a homolog of *pilQ* (*ctsD*, *cj1474c*) [11], shown to encode a pore for the uptake of DNA over the outer membrane into the periplasm, e.g., in *Neisseria gonorrhoeae* [13,14]. The type IV pilus traverses PilQ [15], and DNA binding and retraction of the pilus fiber was shown to move DNA towards the outer membrane, initializing DNA uptake into the periplasm in *Vibrio* [16]. A similar mechanism of “grabbing” external DNA or of DNA binding to a pseudopilus, as proposed for *Neisseria* [17], might also be postulated for *C. jejuni*. 

The periplasmic DNA-binding protein ComE was suggested to be the receptor for incoming DNA in well-studied competent Gram-negative bacteria. ComE was shown to exert force on the DNA molecule, thereby pulling DNA over the outer membrane [18,19]. ComE-mediated import of DNA is a considerably slower process than observed in *H. pylori*, lacking a *comE* homolog and displaying fast pilus-like DNA uptake kinetics [20]. The unique ComH protein, only present in *H. pylori*, was proposed to play a role in the alternative binding of periplasmic DNA [21]. However, ComH accumulation in the periplasm occurred timely after DNA uptake. Thus, the role of this periplasmic protein might be in initialization of DNA uptake and transfer of already imported DNA to the inner membrane ComEC channel. Using ComE for DNA uptake, the binding capacity in the periplasm appeared to be limited to around 40 kb [22] *C. jejuni* harbors the *comE* homolog Cj0011c but knockout mutants were only 10–50-fold impaired in natural transformation activity [23]. The overall DNA uptake process was shown to depend on the presence of a methylated RAATTY motif, excluding foreign DNA from uptake into *C. jejuni* [24]. Upon entry into the periplasm, DNA has to be unzipped while being imported through the inner membrane channel, ComEC, reaching the cytoplasm [10]. A core “competence domain” in ComEC proteins was defined from the analysis of more than 5500 bacteria [25]. Likewise, *C. jejuni* harbors the *comEC* homolog cj1211c, and its knockout led to a transformation-deficient mutant strain still able to take up radiolabeled DNA [26].

Extracellular DNA has been shown to serve as a matrix for biofilm formation and it has been suggested that horizontal gene transfer and biofilm formation are interconnected [27]. For example, in *Streptococcus*, the deletion of the competence protein *comGB* drastically decreased DNA binding and uptake and reduced biofilm formation [28]. In *Clostridioides difficile*, the deletion of *pilJ* and *pilW,* coding for pilin subunits involved in DNA uptake, impaired biofilm formation [29].

In this study, we compared the DNA uptake capacity of *C. jejuni* with that of *Helicobacter pylori*, from which absolute quantitative data are available [30]. Our observations suggested that (i) the median uptake capacity of *C. jejuni* was 30 kb per focus, (ii) the initialization of DNA uptake might be the limiting factor; (iii) the process was finalized within 30 min of contact with DNA; and that (iv) the process stopped upon transport of a finite DNA amount into the periplasm. Mutants lacking key players for outer and inner membrane DNA transport, PilQ, ComE, and ComEC, were constructed, analyzed in the single cell assay, and their negligible role in biofilm formation was observed. Most interestingly, we identified a novel periplasmic pilus protein, Cj0683, with an important role in the initialization of DNA uptake over the outer membrane. 

## 2. Materials and Methods

### 2.1. Strains and Growth Conditions

All strains used in this study are listed in Appendix A. *C. jejuni* 81–176 [31] from −80 °C stock cultures (MAST Group Ltd., Bootle, UK) was grown on Columbia blood agar plates supplemented with 5% defibrinated sheep blood (ColbA, Oxoid, Thermo Fisher Scientific Inc., Waltham, MA, USA) under microaerobic conditions (5% O_2_, 10% CO_2_, rest N_2_; Binder, Tuttlingen, Germany). After subculturing under the same conditions, the strain was grown in a liquid shaking culture in Bolton broth (Oxoid, Thermo Fisher Scientific Inc., Waltham, MA, USA) at 3.5% H_2_, 6% O_2_ and 7% CO_2_ (Air Liquide, Paris, France) and 37 °C using a jar with gas replacement capacity (Oxoid Anaerobia System, Thermo Fisher Scientific Inc., Waltham, MA, USA). For the selection of transformants, ColbA was supplemented with kanamycin (20 µg/mL), apramycin (20 µg/mL), streptomycin (20 µg/mL), and/or chloramphenicol (15 µg/mL) (Sigma Aldrich, Merck KGaA, Darmstadt, Germany). *Helicobacter pylori* J99 [32] was cultured on ColbA or in tryptic soy broth (TSB, Becton Dickinson, Franklin Lakes, NJ, USA) supplemented with 5% fetal calf serum (PAN-Biotech GmbH, Aidenbach, Germany) and 2.5 μg/mL amphotericin B, 0.31 µg/mL polymyxin B, 6.25 µg/mL trimethoprim, and 12.5 µg/mL vancomycin (Sigma Aldrich, Merck KGaA, Darmstadt, Germany). Cells were cultured at 140 rpm under microaerobic conditions (5% O_2_, 10% CO_2_, rest N_2_) and 37 °C.

### 2.2. DNA Uptake Assay, Quantification and Transformation

The DNA uptake assay was performed as described in [8]. In short, genomic *C. jejuni* DNA was extracted using the PureLink Genomic DNA Kit (Thermo Fisher Scientific Inc., Waltham, MA, USA) and fluorescently labelled with Label IT reagent (Mirus Label IT Fluorescein or Cy3, Mirus Bio LLC, Madison, WI, USA) in a volume:weight ratio of 1:1. After pre-culturing bacteria on ColbA and, subsequently, in Bolton broth, *C. jejuni* were subcultured in the same liquid medium but with pH adjustment using 175 µL 1 M sodium hydroxide per 5 mL Bolton broth in order to reach the “ON-status” for natural transformation/competence (pH~7.5 after growth in exponential phase) [8]. *H. pylori* were grown in TSB without pH adjustment. The cultures were incubated at 140 rpm and 37 °C in an atmosphere containing 3.5% H_2_, 6% O_2_, 7% CO_2_ and rest N_2_. After 18 h ± 4 h and generation times of 0.9–1.5 h for *C. jejuni* or 2.6–2.8 h for *H. pylori*, 100–800 µL cells were harvested in exponential growth phase at OD = 0.05–0.6 by centrifugation at ~16,000× *g* for 5 min. The pellet was resuspended in 100 µL BHI supplemented with 100 ng fluorescent genomic *C. jejuni* DNA (1 mg/mL). Cell suspensions were incubated for 30 min under aerobic (*C. jejuni*) or microaerobic (*H. pylori*) conditions (5% O_2_, 10% CO_2_, rest N_2_) at 37 °C before centrifugation at 16,000× *g* for 5 min and resuspension in 15 µL BHI supplemented with 10 U DNase (Roche, Rotkreuz, Schweiz). After the digestion of external DNA for 5 min at 37 °C, cells were immobilized on a 1.5% agarose surface and mounted on an Axio Observer ZI (Zeiss, Jena, Germany) microscope as described previously [8], using 25–100 ms of exposure time. Cells harboring at least one fluorescent focus were considered competent. The ratio of competent cells was calculated. For the quantification of DNA uptake, regions of interest (ROIs) at the DNA uptake locations and in background cells were manually set and mean fluorescence intensity per pixel and number of pixels per ROI were analyzed. The effect of bleaching was checked by subsequent light exposure (4-times 500 ms) to Cy3-labelled λ DNA upon uptake in *H. pylori* (Thermo Fisher Scientific Inc., Waltham, MA, USA). The fourth measurement was 96.1% ± 2.8% compared to the first light exposure of 500 ms, indicating that bleaching was negligible under these conditions.

For the determination of transformation rates, all steps were performed under a microaerobic atmosphere at 5% O_2_, 10% CO_2_, and rest N_2_. Bacterial cells were incubated with unlabeled genomic DNA (gDNA) of a strain carrying a point mutation in *rpsl* (A128G; 81–176-strep). After uptake for 1 h in BHI at 37 °C in the presence of 1 mg/mL gDNA, cells were centrifuged for 5 min at 16,000× *g* and resuspended in fresh BHI with 10 U DNase. Digestion was performed for 1 h at 37 °C. Cells were again centrifuged, resuspended in fresh BHI, and incubated for 3 h at 37 °C before the selection of transformants on ColbA supplemented with 20 µg/mL streptomycin at 37 °C for 2 days. The transformation rate was calculated as the ratio of the number of transformants grown on streptomycin-containing ColbA versus the total number of colonies on non-selective plates. 

### 2.3. Construction of Deletion Mutants and Respective Complemented Strains

Mutants were constructed using site directed mutagenesis (Appendix A). A nonpolar kanamycin cassette [33] was fused with about 500 bp homologous adjacent regions to recombine into *pilQ* (*ctsD = cj1474c*), *comE* (*cj0011c*) or *comEC* (*cj1211*). For the deletion of *cj0683,* an apramycin cassette was used instead [34] to further allow the construction of double mutants. Hence, 5′ flanking regions were amplified using primer pair C29/C30 for *comEC,* C34/C35 for *pilQ*, C44/C45 for *comE*, and C98/C99 for *cj0683* (Appendix A). Cloning primers were purchased as HPLC-grade, PCR verification primers were desalted (Sigma Aldrich, Merck KGaA, Darmstadt, Germany). Primers C30, C35, C45 contained a 15 bp region complementary to the 5′end of the kanamycin cassette. Primers C98/C99 harbored an *Eco*RI site and 29 bp complementary to the 5′ end of the apramycin cassette. Furthermore, 3′ flanking regions were amplified using C31/C32 for *comEC,* C36/C37 for *pilQ*, C46/C47 for *comE* and C100/C101 for *cj0683*. Primers C31, C36, and C46 harbored a 24-bp region complementary to the 3′ end of the kanamycin cassette. Primer C100/C101 contained an *Eco*RI site and 14 bp complementary to the 3′ end of the apramycin cassette. For each PCR genomic DNA of NCTC 11168 [35] was used as a template. The kanamycin resistance gene was amplified from pUC18K2 with H13/H14. Primers H11/H12 were used to amplify the apramycin cassette from pUC1813*apra* [34]. Using these complementary regions, the kanamycin gene was fused with the flanking regions of *pilQ*, *comE*, and *comEC* by PCR. Accordingly, the apramycin gene was fused with the flanking regions of *cj0683*. The fusion fragments for the deletion of *pilQ*, *comE*, and *comEC* were cloned into a pCR2.1-TOPO vector and transformed into One Shot TOP10 chemically competent *Escherichia coli* (Thermo Fisher Scientific Inc., Waltham, MA, USA) according to the manufacturer’s protocol. For the deletion of *cj0683*, the fusion fragment was methylated by *Eco*RI methyltransferase (New England Biolabs, Ipswich, MA, USA) for 1 h at 37 °C followed by heat inactivation at 65 °C for 15 min.

The vectors were purified using the GeneJet Plasmid Mini Prep kit (Thermo Fisher Scientific Inc., Waltham, MA, USA) and transferred into strain NCTC 11168 via electroporation modified according to Van Vliet et al. 1998 [36]. Note that electroporation was the method of choice for construction of *C. jejuni* mutants before the discovery of the methylation motif, essential for DNA uptake [24]. The introduction of DNA into *C. jejuni* by electroporation, however, is much less efficient than natural transformation. Thus, in this study, only the three mutants, Δ*pilQ*, Δ*comEC*, and Δ*comE*, were constructed by electroporation of the pCR2.1-TOPO vectors into NCTC 11168 and 100 ng gDNA of those strains was used for transfer of the respective target deletion to strain 81–176 as described in Section 2.2. The complementation mutants and the Δ*cj0683* mutant were constructed by direct natural transformation of the fusion fragment or the complementation vectors (see below) into strain 81–176. For this purpose, *Eco*RI restriction sites were introduced into the constructs and methylated as mentioned above so that the constructs harbored a proper signal for DNA uptake during natural transformation. For construction of the double mutant Δ*cj0683*Δ*comE,* 100 ng of the methylated fusion PCR product of Δ*cj0683* was transformed into 81–176Δ*comE* mutant. Transformants were selected on kanamycin and apramycin. Deletion of *pilQ*, *comE*, *comEC*, or *cj0683* in 81–176 and insertion of the nonpolar kanamycin cassette or apramycin cassette was verified by PCR (Appendix A).

Complementation of genes was performed by the introduction of the respective wild type gene in the *hsdM* (*cj1553c*) locus of 81–176 using the suicide vector pSB3021 as described previously [37] (Appendix A). Since the deletion of *pilQ, comEC*, and *cj0683* abolished or drastically reduced natural transformation, we first introduced a second copy of *pilQ, comEC*, and *cj0683* before deletion of those genes at their native locations. pSB3021 contains a chloramphenicol acetyltransferase gene and the target gene is co-expressed with the resistance cassette. In detail, the complete *pilQ* was amplified with C113/C114, *comEC* with C115/C116, and *cj0683* with C117/C118, generating each at the 5′ end a 20 bp Shine-Dalgarno region from *pglF*, preceded by an *Eco*RI site for methylation and an *Nco*I-site at the very 5′ end for cloning. Likewise, a second *Eco*RI site followed by an external *Xho*I-site was introduced at the 3′ end. PCR products and the vector were digested using *Nco*I and *Xho*I and the respective insert was ligated into pSB3021. After transformation of the ligation product into One Shot TOP10 chemically competent *E. coli,* the resulting vectors were methylated as described above and 100 ng of each vector was transformed into 81–176. The transformants were selected with 15 µg/mL chloramphenicol. After confirmation of the correct insertion of the gene construct in *hsdM*, the strains were then transformed with genomic DNA of 81–176Δ*pilQ,* 81–176Δ*comEC*, or 81–176Δ*cj0683* and selected with both chloramphenicol and kanamycin or chloramphenicol and apramycin. The strains were verified by PCR (Appendix A).

### 2.4. Biofilm Formation Assay

*C. jejuni* was grown on Mueller-Hinton-Agar supplemented with 5% sheep blood (MHAB; Oxoid, Thermo Fisher Scientific Inc., Waltham, MA, USA) for 72 h under microaerobic conditions (6% O_2_, 10% CO_2_, rest% N_2_) created with the Mart Anoxomat system (Mart Microbiology BV, Advanced Instruments, Norwood, MA, USA). For liquid cultures, colonies were transferred into Brucella broth (Becton Dickinson, Franklin Lakes, NJ, USA) and incubated under the microaerobic conditions mentioned above for 24 h. The control strain *Pseudomonas aeruginosa* DSM 1117 (DSMZ strain collection, Braunschweig, Germany) was cultured on BHI agar (Merck KGaA, Darmstadt, Germany) for 24 h and 37 °C or in a liquid shaking culture in Luria Bertani (Merck KGaA, Darmstadt, Germany) at 37 °C.

The biofilm assays were performed according to the method described before with slight modifications [38]. Briefly, for generating a 100-fold dilution of the inoculum, the respective amount of overnight culture was transferred into fresh Mueller-Hinton broth (MHB; Oxoid, Thermo Fisher Scientific Inc., Waltham, MA, USA) to achieve a cell count of approximately 1 × 10^7^ CFU/mL, which was checked by drop plating on MHAB. The outer wells of a sterile uncoated 96-well polystyrene microtiter plates (VWR, Dresden, Germany) were filled with 250 µL sterile medium to minimize evaporation in the central wells. For each strain, six wells were inoculated with 100 µL of the respective cell suspension and sterile MHB was used as negative control in one column of 6 wells. The plates were incubated at 37 °C for 72 h under microaerobic conditions, using CampyGen packs (Oxoid, Thermo Fisher Scientific Inc., Waltham, MA, USA) in an anaerobic jar. In order to determine the biofilm mass, the supernatant was carefully removed, and the wells were washed once with 150 µL PBS (Merck KGaA, Darmstadt, Germany) to remove unbound bacterial cells. Afterwards, the plates were dried for 30 min at 60 °C before the addition of 150 µL 0.1% crystal violet (CV) solution (Sigma-Aldrich, Merck KGaA, Darmstadt, Germany) to each well and incubation for 30 min at room temperature. To remove excess CV, the wells were washed with 400 µL and once again with 150 µL PBS. After drying for 30 min at room temperature, bound CV was dissolved in 150 µL of 96% ethanol for 15 min. Hence, 100 µL of the supernatant was transferred to a new microtiter plate and the absorbance was measured at 595 nm using a microplate reader (EL800, Aglient BioTek, Santa Clara, CA, USA). For final analysis, the median absorbance from the respective negative controls was subtracted from each value of the corresponding samples.

### 2.5. Statistical and Bioinformatic Analysis

Statistical significance was tested using a Mann-Whitney U-test in R version 4.2.2. BLAST analysis was performed with the online tool of the National Center for Biotechnology Information (NCBI; https://blast.ncbi.nlm.nih.gov; last accessed on 9 December 2022). For the calculation of protein secretion signals, Phobius Webtool was used (https://phobius.sbc.su.se/; Stockholm Bioinformatics Center, Sweden; last accessed on 9 December 2022). Calculation of theoretical pI was performed with Expasy (https://www.expasy.org/resources/compute-pi-mw; last accessed on 9 December 2022).

## 3. Results

### 3.1. Multiple Fluorescent DNA Molecules Are Located within Distinct Periplasmic Foci during C. jejuni DNA Uptake 

*C. jejuni* can enter a competence phase, in which nearly half of the cells were capable to take up external DNA at one to four distinct locations within a cell [8]. Here, we wanted to know how much DNA was imported at distinct DNA uptake locations and if multiple DNA molecules were imported at one location. For this purpose, we used our established single cell assay to monitor fluorescent DNA uptake in *C. jejuni* [8]. In order to distinguish different DNA molecules, we prepared two batches of either fluorescein or Cy3 labelled genomic *C. jejuni* DNA. First, we checked whether *C. jejuni* DNA uptake was independent of the fluorophore. Indeed, nearly the same number of cells took up either fluorescein or Cy3 labelled DNA, if differently labelled DNA batches were added to separate cell suspensions for 30 min (35.3% ± 2.4% for fluorescein DNA or 36.4 ± 3.0% for Cy3 DNA, Figure 1A, green and orange bars). 

When *C. jejuni* was incubated with both fluorescein and Cy3 DNA in parallel, we observed that 16.4% ± 3.6% of all cells with active DNA uptake imported at least two DNA molecules at the same location, since both fluorescein DNA and Cy3 DNA were detected in a single focus (Figure 1B, yellow bar). In addition, 3.5% ± 1.1% of competent cells displayed two separate foci with either fluorescein or Cy3 DNA (Figure 1B, green/orange hatched bar). Again, the fraction of cells that solely took up either fluorescein labelled DNA or Cy3 DNA was similar (40.7% ± 5.7% for fluorescein and 39.4 ± 6.0% for Cy3 DNA).

To verify the location of imported fluorescent DNA in the periplams of *C. jejuni*, we checked the transformation rate using either native unmodified DNA or covalently labelled substrates of a streptomycin resistant strain carrying the A128G point mutation in *rpsL*. Using unmodified DNA as substrate, transformation rates of 6.6 × 10^−4^ ± 6.4 × 10^−4^ were observed. In contrast, fluorescein and Cy3 labelled DNA were very poor substrates for transformation in *C. jejuni*, since transformation rates were nearly at the detection limit of 8.5 × 10^−9^ ± 2.4 × 10^−9^. The results can be explained by the fact that covalently labelled DNA is situated in the periplasm because of steric hindrance of the covalent fluorophore, inhibiting further transport into the cytoplasm as previously suggested for *H. pylori* [20]. Very rare events of transformation might be due to very few unlabelled molecules present in the labelled fraction of DNA. We showed before that the fraction of cells importing covalently modified DNA and transformation rates with native substrate correlated well [8], indicating the suitability of the single cell assay for the analysis of natural transformation. The periplasmic trapping of covalently modified DNA is a further advantage of the assay for the quantification of total DNA uptake capacity (see below).

### 3.2. Estimation of DNA Uptake Capacity of C. jejuni Reveals Imported DNA Amounts in Median of around 30 kb

*C. jejuni* was able to take up at least two DNA molecules at one distinct location. Previously, we quantified the amount of DNA uptake per focus and per cell in *H. pylori* [30]. Quantification was performed by analyzing the uptake of Cy3 labelled DNA of the 48.5 kb bacteriophage λ and normalization of fluorescence intensity to single λ molecules of the same labelling batch. For *H. pylori* J99, a median of 108 kb per DNA focus and 351 kb per cell were observed [30]. *C. jejuni* only takes up native genomic DNA methylated at RAATTY sites, but no foreign λ DNA. Genomic *C. jejuni* DNA contains different fragment sizes and sequences. Thus, single molecule fluorescence intensity determination is not suitable. However, on average, the length of genomic *C. jejuni* DNA was ≥20 kb as estimated by gel electrophoresis. Thus, a similar uptake pattern should be expected for *H. pylori* in the presence of genomic instead of λ DNA, since unlike *C. jejuni*, the bacterium is indifferent towards the DNA source. In order to approximate the DNA capacity of *C. jejuni*, we challenged both *C. jejuni* and *H. pylori* cells in parallel with the same batch of freshly labelled genomic *C. jejuni* DNA (Figure 2). 

In *C. jejuni*, the fraction of competent cells increased nearly two-fold from 22.7% ± 3.2% after 10 min to 42.2% ± 7.5% within 2 h of incubation with fluorescent DNA (Figure 2A). In addition, the number of foci per cell increased with time of contact with labelled DNA in *C. jejuni* (Figure 2B). In comparison, 10-min incubation resulted in 73.9% ± 3.1% of *H. pylori* with imported fluorescent DNA, from which more than half of the bacteria harbored multiple active DNA uptake locations.

In order to compare the quantitative amount of DNA in distinct uptake locations, we analyzed the total fluorescence intensity of at least 450 DNA foci per condition (Figure 3). For this purpose, the mean fluorescence intensity and the area of regions of interest (ROI) were exported and background values obtained were subtracted from bacteria without DNA uptake. The median fluorescence intensity per focus, i.e., the amount of imported DNA, was 3.7-fold ± 0.5 higher in *H. pylori* compared to *C. jejuni* (Figure 3). 

This was even “visible to the naked eye”, since for imaging within the limit of the 12-bit camera (below saturation; 4096 grey values), a four-fold lower exposure time had to be used for *H. pylori* than for *C. jejuni* to avoid the saturation of grey values. We observed that the distribution of the DNA amount per focus was relatively constant between 10 min and 2 h of uptake time in *C. jejuni* (Figure 3B). This might indicate that the maximal DNA amount per distinct uptake location was limited. A similar distribution of fluorescence intensities per DNA focus was observed for *H. pylori* (Figure 3C), with an approximate four-fold shift in absolute fluorescence intensities (Figure 3A,C).

### 3.3. Essential Roles of pilQ (cj1474c) and comEC (cj1211) but Negligible Contribution of comE (cj0011c) to DNA Uptake in C. jejuni

To gain further insights into the DNA uptake process, we created knock-out mutants by the insertion of a non-polar kanamycin cassette and deletion of genes with roles in natural transformation. The mutants lacked one of the following proteins. The protein Cj1474c has distant homology (25% identity, 43% similarity) to the outer membrane type IV pilus secretin PilQ of *N. gonorrhoeae* and to GspD (25% identity, 46% similarity) of *B. subtilis*, which is part of the type II secretion system secretin. Cj0011c shares 54% identity (71% similarity) with ComE of *N. gonorrhoeae* and 59% identity (83% similarity) with ComEA of *B. subtilis*. ComE is known as the periplasmic DNA-binding protein, generating the force for DNA uptake over the outer membrane [18,19]. Furthermore, Cj1211 shows homology to *H. pylori* ComEC protein (34% identity; 56% similarity).

The 81–176Δ*pilQ*, lacking the putative outer membrane pore, was completely deficient of DNA uptake and, consequently, natural transformation (Figure 4). This means that we did not observe a single DNA uptake focus within ≥10,000 analyzed bacteria, nor any transformant within the detection limit of 8.5 × 10^−9^ ± 2.4 × 10^−9^.

Δ*comE* displayed a marginally reduced fraction of bacteria with active DNA uptake of 24.6% ± 5.5% (*n* = 12), while the transformation rate was reduced by 1.3 log_10_ relative to the wild type, indicating a non-essential role of ComE in natural transformation of *C. jejuni*. As expected for a role in the inner membrane, not disturbing DNA uptake into the periplasm, a similar fraction of cells of 26.2% ± 4.9% (*n* = 12) showed active DNA uptake in Δ*comEC* compared to the wild type. Furthermore, the Δ*comEC* mutant was transformation deficient (Figure 4B), supporting its role as an inner membrane channel. 

The two mutants with relevant defects in natural transformation, Δ*pilQ* and Δ*comEC*, were genetically complemented. Both mutants showed natural transformation rates, which were 1.5–2 log_10_ lower than the wild type (Figure 4B). Hence, partial complementation was observed. The lower fraction of cells with active DNA uptake harbored DNA foci with similar fluorescent intensity to the wild type, suggesting the full functionality of these DNA uptake complexes (Appendix A). In theory, there is a possibility of re-integration of the *pilQ* gene from *hsdM* into its native location in a subpopulation of cells, putatively explaining the low fraction of competent cells of Δ*pilQ*-compl. In order to exclude this possibility, we selected Δ*pilQ*-compl-strep clones which had successfully been transformed with the streptomycin marker and checked that those cells displayed the same phenotype as the parental strain (Appendix A). 

### 3.4. Active DNA Transport Capacity Has No Impact in Biofilm Formation of C. jejuni

Since extracellular DNA is a well-known matrix for biofilm formation, we wondered if the DNA uptake complex is important for the binding of matrix DNA necessary for biofilm formation in *C. jejuni*. Therefore, our mutants deficient or hampered in DNA uptake and/or natural transformation were tested for their capacity to produce biofilms. Biofilm formation of the mutant strains Δ*pilQ*, Δ*pilQ*-compl, Δ*comE*, Δ*comEC,* and Δ*comEC*-compl was generally low but similar to the wild type strain (Figure 5). As a control, we used *Pseudomonas aeruginosa* DSM 1117, which produced on average approximately 6.5-fold more biomass attached to the abiotic surface than *C. jejuni*. From these experiments, we concluded that biofilm formation in *C. jejuni* is generally weak and independent of active DNA uptake.

### 3.5. Cj0683 Is a Novel Periplasmic Protein Essential for Efficient Initialization of DNA Uptake in C. jejuni

A mutant lacking the periplasmic DNA-binding protein ComE, involved in DNA uptake in most other Gram-negative bacteria, showed nearly wild type DNA uptake capacity and natural transformation in *C. jejuni* (Figure 4). In addition, BLAST analysis revealed the absence of a *comE* homolog in *Campylobacter* spp. except for *C. jejuni* and *C. cuniculorum* isolated from rabbits. ComE unspecifically binds to DNA in the periplasm, generating force on the macromolecule, which in turn leads to the import of DNA. ComE is a small protein, harboring a classical N-terminus for secretion into the periplasm and an alkaline isoelectric point, rendering the protein positively charged at pH 7. 

Hence, we initially wondered whether another periplasmic protein could take over the function of ComE in *C. jejuni*. Therefore, we searched for putative periplasmic proteins in *C. jejuni* by screening annotated Genebank files and verifying the putative secretion signal using Phobius Webtool. The putative periplasmic protein Cj0683 has similar properties as ComE. A secretion signal calculated using the Phobius Webtool predicted amino acid residues 27–145 to be secreted from the cytoplasm. The N-terminus additionally harbors a motif, including a phenylalanine residue predicted to be methylated during pilin biogenesis. The theoretical pI of the mature protein was calculated to be 8.3. BLAST analysis revealed that Cj0683 is unique for *Campylobacter* spp., since no homolog was found in other bacteria. Gene context analysis showed that *cj0683* is situated in an operon preceding the gene *priA* in *C. jejuni*, *C. coli* and *C. lari*. PriA had been shown to be involved in natural transformation in *Neisseria*, putatively acting as helicase for the import of DNA from the periplasm into the cytoplasm [39]. In addition, Cj0683 was shown to interact in a yeast two-hybrid system with CtsE [40], a protein with a nucleotide binding motif, essential for natural transformation of *C. jejuni* [12]. Thus, Cj0683 was a good candidate for implication in DNA uptake in *C. jejuni*. 

For this reason, a deletion mutant was constructed by the insertion of a non-polar apramycin cassette into *cj0683*. In addition, a complemented mutant was constructed by the first introduction of a second copy of *cj0683* in the *hsdM* locus and subsequent deletion of *cj0683* in its native locus. As performed for the other mutants, Δ*cj0683* and the complemented mutant Δ*cj0683*-compl were incubated with fluorescent DNA for 30 min. Interestingly, the Δ*cj0683* mutant, lacking the unique periplasmic protein, was nearly completely deficient of DNA uptake, as expected from its in silico characteristics. Only a minor fraction of 0.09% ± 0.13% (cumulative results of both fluorophores; *n* = 10) of the cells displayed a DNA focus, and the transformation rate was accordingly reduced by more than 3.7 log_10_ (Figure 6). 

To exclude that these rare events were due to suppression mutations in Δ*cj0683*, we transformed the Δ*cj0683* with the streptomycin resistance marker and selected the rare streptomycin-resistant transformants (Δ*cj0683*-strep). If suppression mutations had occurred in the bacteria displaying functional DNA uptake, we expected a wild type phenotype of DNA uptake in the Δ*cj0683*-strep clones. Indeed, we identified similar rare events (0.04% ± 0.03%; cumulative results; *n* = 9) as observed in the parental Δ*cj0683* strain (Appendix A). 

We further wondered whether ComE can partially take over the function of Cj0683 and whether the remaining DNA uptake and transformation rate of the Δ*cj0683* mutant was dependent on ComE. Therefore, we constructed a double mutant Δ*cj0683*Δ*comE*. A comparable fraction of 0.11% ± 0.21% of bacteria displayed functional DNA uptake in the double mutant, suggesting that ComE was not involved in these rare events (Figure 6A). Δ*cj0683*Δ*comE*-strep clones obtained after the transformation of the streptomycin resistance marker maintained the same phenotype of rare events of DNA uptake as observed in the parental strain (Appendix A). However, as already observed for the Δ*comE* mutant relative to the wild type, the transformation rate in the double mutant was slightly (five-fold) reduced compared to the single Δ*cj0683* mutant (Figure 6B), hinting at a minor downstream role of ComE in either DNA protection and/or delivery to the inner membrane channel. 

We further quantified the DNA uptake capacity of rare events observed in Δ*cj0683* and Δ*cj0683*Δ*comE* compared to the wild type (Figure 7). A maximal two-fold reduction of median imported DNA amounts was observed in rare events, with some outliers harboring similar or even higher DNA amounts per focus compared to the wild type. Hence, we conclude that the rare DNA uptake events were caused by fully functional DNA uptake over the outer membrane. The absence of Cj0683 led to drastically decreased DNA uptake probability, consistent with a putative role of Cj0683 as part of the (pseudo-) pilus involved in the acquisition and initiation of DNA uptake but not in force-generation as proposed for ComE in other bacteria. 

## 4. Discussion

### 4.1. DNA Uptake Capacity in C. jejuni Is around Four-Fold Lower than in H. pylori 

We previously observed that the fraction of cells displaying at least one active DNA uptake location correlated well with transformation rate using a point mutation resistance marker [8]. By challenging *C. jejuni* cells with fluorescein and Cy3 labelled DNA, we showed that (i) cells were able to take up both substrates at one visually distinct location and (ii) multiple locations of DNA uptake were present per cell (Figure 1). 

*C. jejuni* only imports DNA which is methylated by the CtsM methylase at the N6-methyladenine residue of the RA^m6^ATTY motif [24]. This is why the 48.5 kb DNA of bacteriophage λ, which was used for absolute quantification in *H. pylori* [30], was not a suitable substrate for *C. jejuni*. Instead, genomic *C. jejuni* DNA is imported with molecule sizes of ≥20 kb. Thus, we incubated *C. jejuni* and *H. pylori* with the same batch of Cy3 labelled *C. jejuni* DNA in parallel for the relative estimation of DNA capacity in *C. jejuni* compared to *H. pylori*. 

Three main results were obtained. The first result was that the fraction of cells with imported fluorescent foci and with multiple DNA uptake locations per cell increased by around two-fold between 10 min and 2 h of incubation of *C. jejuni* with DNA. Second, the amount of DNA per focus increased slightly from 10 to 30 min but was quite stable in time. We can exclude that competence was developed during the 2 h of incubation, since we used aerobic conditions, under which DNA transport is unaffected but competence development is suppressed [8]. Thus, these first two observations suggested that (i) the initialization of a DNA uptake event might be the limiting factor, that (ii), once the DNA process started, the molecule can be incorporated within 10 min (process completed after 30 min), and that (iii) the process stops upon transport of a finite DNA amount into the periplasm. The third main result revealed that the median fluorescence intensity was 3.7-fold lower than for *H. pylori* J99. *H. pylori* J99 imported a median DNA amount of 108 kb per focus, i.e., per visually resolved uptake location [30]. Assuming that *H. pylori* does not differentiate between λ DNA and genomic *C. jejuni* DNA, we estimate that *C. jejuni* imported a median amount of 30 kb DNA per focus. In a few DNA uptake locations (outliers in Figure 3), 5–6-fold more fluorescent DNA was detected. Intriguingly, the distribution of DNA amounts in *H. pylori* uptake locations was quite similar (Figure 3C) but shifted up by ~4-fold relative to the *C. jejuni* distribution with also very few outliers, harboring an excess of DNA. Hence, we conclude that those rare events, leading to outliers, reflected multiple DNA uptake complexes, by chance situated in unresolvable proximity. 

### 4.2. The Periplasmic Protein ComE Is Negligible for DNA Uptake in C. jejuni While ComEC and PilQ Play Essential Roles in Natural Tansformation 

We showed that a mutant lacking the putative outer membrane pore PilQ was completely deficient of DNA uptake and natural transformation (Figure 4A,B). This confirmed its essential role for DNA uptake over the outer membrane. The Δ*comEC* mutant was completely transformation deficient but took up DNA into the periplasm (Figure 4A,B), being consistent with its proposed essential role as an inner membrane channel. 

Complementation of Δ*comEC* and Δ*pilQ* led to the partial recovery of transformation rates (1.5 log_10_ and 2 log_10_ reduced compared to the wild type, respectively). The Δ*pilQ*-compl strain was also similarly restored concerning the fraction of competent cells (1.5 log_10_ reduction compared to the wild type). Partial and not full complementation is expected if stoichiometry of membrane proteins is essential for functional complex formation. This might be the case for our complemented genes, which were under artificial control at a different genetic location. Partial complementation of *ctsX* and *ctsP*, involved in natural transformation, was also observed previously [11]. 

*N. gonorrhoeae* was shown to maximally take up 40 kb into the periplasm [22]. This limitation was attributed to the finite binding capacity of the periplasmic protein ComE. Interestingly, in *C. coli* and *C. lari,* close relatives of *C. jejuni*, a *comE* homolog is absent from the genomes, although both species are naturally competent and harbor homologs of *comEC* and *pilQ*. We created a Δ*comE* knock-out mutant in *C. jejuni*. As observed before [23], transformation levels were slightly reduced by 1.3 log_10_. In our single cell assay, uptake of DNA into the periplasm was comparable to the *comEC* mutant. Thus, the outer membrane transport of DNA was nearly unaffected compared to the wild type. These results suggested that ComE does not play an important role in the DNA uptake of *C. jejuni*, particularly not as a driving force for the uptake of the macromolecule over the outer membrane, as shown for *N. gonorrhoeae* or *Vibrio cholera* [18,19]. However, the slight reduction in the transformability of the Δ*comE* mutant might hint at putative downstream effects of ComE in *C. jejuni*, e.g., the protection of incoming DNA. 

### 4.3. DNA Uptake Does Not Influence Biofilm Formation in C. jejuni

In other bacteria, the lack of proteins involved in DNA uptake was shown to impact natural transformation and biofilm formation [28,29]. However, for *C. jejuni*, we did not observe that the capacity to take up DNA from the environment has any effect on biofilm formation under our conditions (Figure 5). The statistically significant difference between wild type and the complemented strains has no biological relevance since there were no significant differences between wild type and mutants. Generally, biofilm formation in *C. jejuni* was weak, which is in-line with previous data. Using a similar assay, Reeser and colleagues showed comparable levels of biofilm formation in *C. jejuni*, with absorption levels after staining with crystal violet of ≤0.2 [38]. In addition, a recent study showed that *C. jejuni* biofilm formation was increased in the presence of already existing *Pseudomonas* biofilms, in particular under flow conditions [41]. This might hint at the fact that initial attachment to an abiotic surface is challenging for *C. jejuni*.

### 4.4. The Unique Periplasmic Protein Cj0683 Is Essential for Efficient Initialization of DNA Import into the Periplasm

The periplasmic DNA-binding protein ComE, mediating DNA uptake over the outer membrane in well-studied competent Gram-negative bacteria [18,19], apparently did not play a major role for transformation in *C. jejuni* (Figure 4). 

We tried to find an alternative periplasmic protein in *C. jejuni*, suitable for DNA binding and important for the process of DNA uptake. Hence, with Cj0683, we deciphered a good candidate, since (i) the protein seemed positively charged and able to bind DNA at physiological pH, (ii) it harbored a secretion signal to be exported into the periplasm, (iii) the N-terminus of Cj0683 shared homology with type IV pilus proteins, and (iv) the protein interacted in a yeast two-hybrid assay with CtsE, a soluble protein with a nucleotide binding site shown to be important for natural transformation in *C. jejuni* [12].

Indeed, a Δ*cj0683* mutant showed a near absence of DNA uptake. Residual DNA uptake capacity was independent of ComE presence, since the double mutant Δ*cj0683*Δ*comE* displayed the same phenotype of rare DNA uptake events. Our experimental approach also excluded suppression mutations in these bacteria with functional DNA uptake. 

Which role might be attributable to Cj0683? We considered in detail the rare events of DNA uptake in the Δ*cj0683* and the Δ*cj0683*Δ*comE* mutants. We observed similar median DNA amounts per uptake location in both mutants as compared to the wild type (maximally two-fold less, Figure 7). These results suggest that the probability of DNA import was drastically reduced, while subsequent DNA uptake into the periplasm was unaffected. Hence, Cj0683 probably does not play the same role as ComE (force generation) or as ComH (transfer of already imported DNA to the ComEC channel) [21]. In contrast, it is tentative to speculate that Cj0683 might be part of the competence (pseudo)-pilus, grabbing external DNA, thereby initializing DNA import events over the outer membrane in *C. jejuni*. This hypothesis would also be consistent with Cj0683 N-terminal homology to pilus proteins. 

## 5. Conclusions

*C. jejuni* is an important food-borne pathogen with a high capacity for genetic diversity via natural transformation. The unravelling of the mechanism of DNA uptake is important to reveal unique proteins important for this process for targeted control strategies. Our results showed the high DNA uptake capacity of *C. jejuni* on a single cell level and we identified a novel competence protein, Cj0683, with important implications in the number of DNA uptake events over the outer membrane.

## Figures and Tables

**Figure 1 biomolecules-13-00514-f001:**
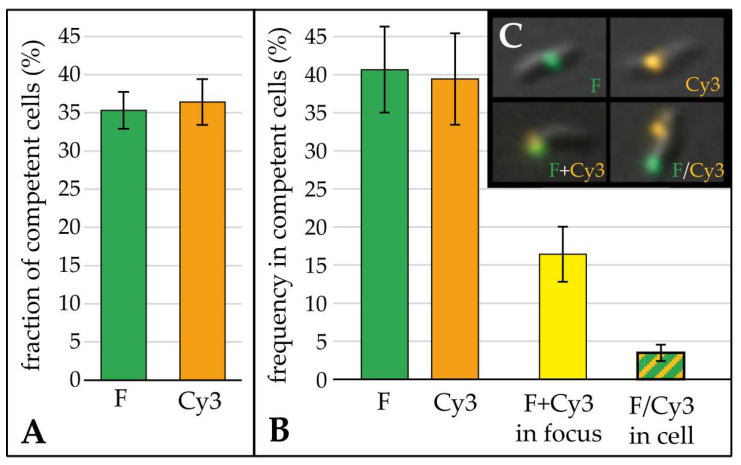
Visualization of import of multiple DNA molecules within one DNA uptake location. Fluorescein (F) and Cy3 labelled DNA was either added for 30 min to separate *C. jejuni* 81–176 suspensions ((**A**), *n* ≥ 5) or in parallel to one bacterial competent culture ((**B**,**C**), *n* = 4) and the fraction of cells with F and/or Cy3 fluorescent foci were analyzed. In (**A**) the fraction of active cells relative to the total number of bacteria is shown, in (**B**) the fraction of cells with fluorescent foci relative to the overall fraction of competent cells is depicted. Green bars, cells with only fluorescein labelled DNA foci; orange bars, cells with only Cy3 labelled DNA foci; yellow bar, cells with both fluorescein and Cy3 DNA in one/or two single focus/foci; green/orange hatched bar, cells with at least two separate foci, with either fluorescein or Cy3 DNA. (**C**), overlay image of differential phase contrast (DIC), Cy3 and F channel. Green, fluorescein focus; orange, Cy3 focus.

**Figure 2 biomolecules-13-00514-f002:**
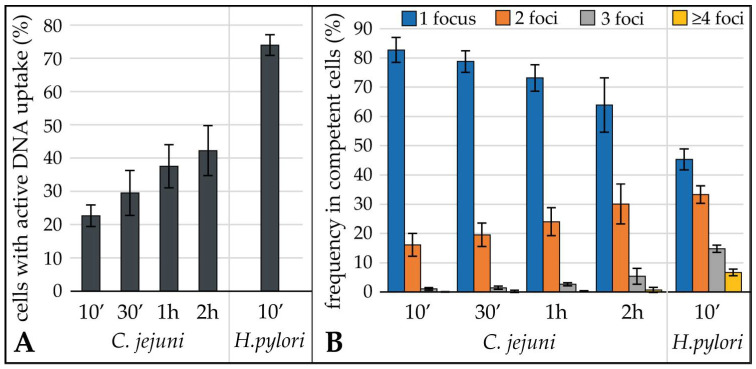
The fraction of competent *C. jejuni* and of DNA uptake locations per cell increased in time of contact with fluorescent DNA. Competent *C. jejuni* 81–176 were exposed to fluorescent DNA for different time periods and the fraction of cells with active DNA uptake was monitored in comparison to 10 min DNA uptake in *H. pylori* J99 (**A**). Proportion of cells with distinct number of DNA uptake locations (**B**). In (**B**), fraction of cells relative to total amount of competent cells are depicted. Mean and standard deviations were derived from five independent experiments.

**Figure 3 biomolecules-13-00514-f003:**
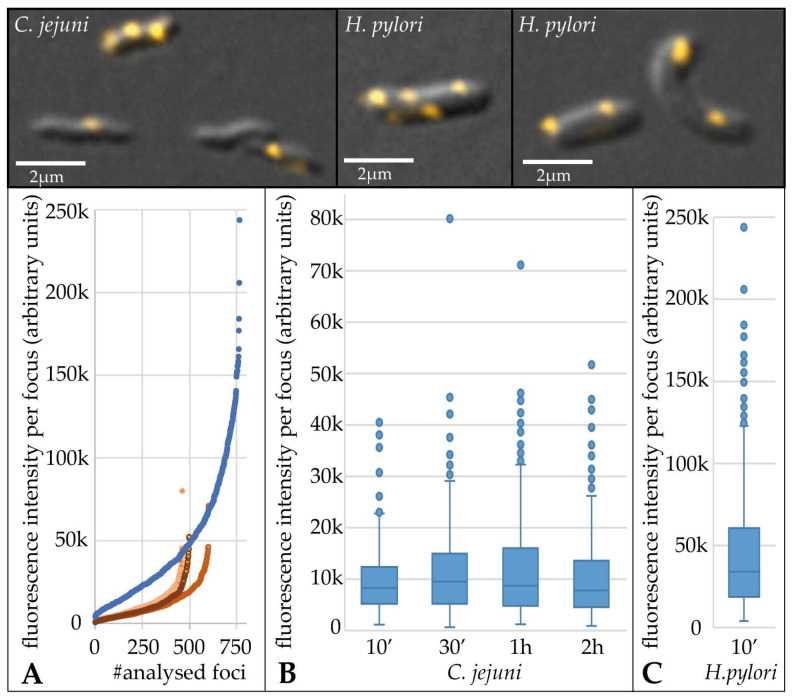
The median amount of DNA within single foci of *C. jejuni* was constant over time and ~4-fold less than in *H. pylori*. *C. jejuni* 81–176 and *H. pylori* J99 were incubated with the same batch of fluorescent genomic *C. jejuni* DNA labelled with Cy3. Uptake was followed in time for *C. jejuni* and compared to 10 min uptake in *H. pylori*. (**A**) Analysis of DNA foci in *C. jejuni* (red; light to dark corresponds to incubation times) and in *H. pylori* (blue), sorted according to fluorescence intensity. (**B**) (*C. jejuni*) and (**C**) (*H. pylori*) distribution of fluorescence intensities as boxplots; the boxplot length corresponds to the interquartile range (IQR; 50%) of data, the horizontal bar indicates the median value; whiskers represent 1.5 × IQR or the maximum/minimum value of the dataset; dots, outliers. Upper panel, merged DIC and Cy3-channel example images of *C. jejuni* and *H. pylori* cells after import of DNA. Note that Cy3 exposure times were 25 ms for *H. pylori* and 100 ms for *C. jejuni*. Scale bar of 2 µm. One representative experiment (out of three) is shown.

**Figure 4 biomolecules-13-00514-f004:**
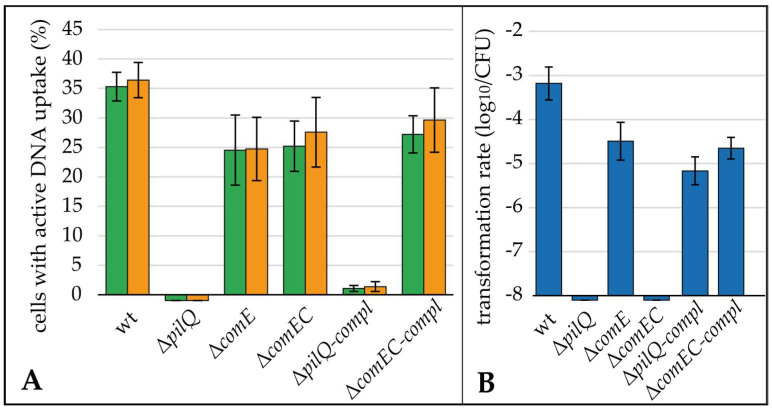
DNA uptake and transformation rate in 81–176 (wt), the Δ*pilQ*, Δ*comE*, Δ*comEC* mutants and the complemented strains Δ*pilQ-compl* and Δ*comEC-compl*. (**A**) Cells were incubated with fluorescein (green bars) or Cy3 labelled (orange bars) DNA of strain 81–176 and the fraction of cells with at least one active uptake location was counted; (**B**) Transformation rate was determined in parallel using a streptomycin marker. Mean and standard deviations are shown from ≥4 independent experiments.

**Figure 5 biomolecules-13-00514-f005:**
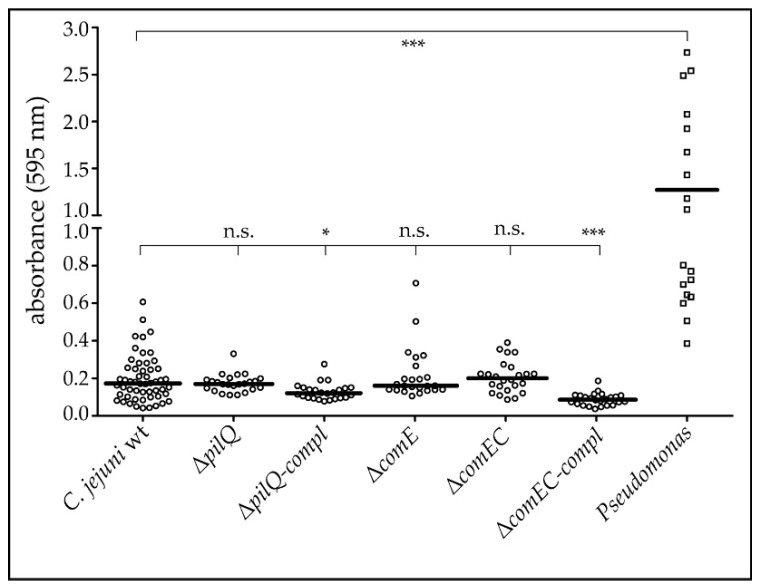
Biofilm formation is independent of active DNA transport or natural transformation in *C. jejuni*. Upon growth in microtiter plates for 72 h and suitable washing of unbound bacteria, crystal violet staining indicated the intensity of biofilm formation, measured as absorbance at 595 nm. *C. jejuni* 81–176 (wt) and its isogenic mutants Δ*pilQ*, Δ*comE*, Δ*comEC* and the respective complemented strains were tested, using *Pseudomonas aeruginosa* as reference for a typical biofilm forming bacterium. Horizontal bar, mean of ≥3 independent experiments; n.s., not significant; *, *p* < 0.05; ***, *p* < 0.001.

**Figure 6 biomolecules-13-00514-f006:**
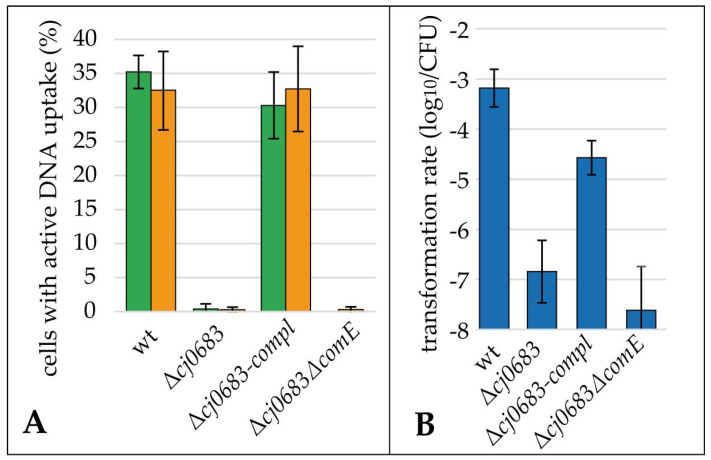
Cj0683 plays a pivotal role for DNA uptake into the periplasm of *C. jejuni*. DNA uptake (**A**) and transformation rate (**B**) in 81–176 (wt) and the mutant strains Δ*cj0683,* Δ*cj0683-compl* and Δ*cj0683*Δ*comE*. (**A**) Cells were either incubated with fluorescein- (green bars, except for Δ*cj0683*Δ*comE*) or Cy3-labelled (orange bars) genomic DNA of 81–176 for 30 min. (**B**) Transformation rate was determined using unlabeled genomic DNA containing streptomycin resistance. Error bars indicate standard deviation (*n* ≥ 4).

**Figure 7 biomolecules-13-00514-f007:**
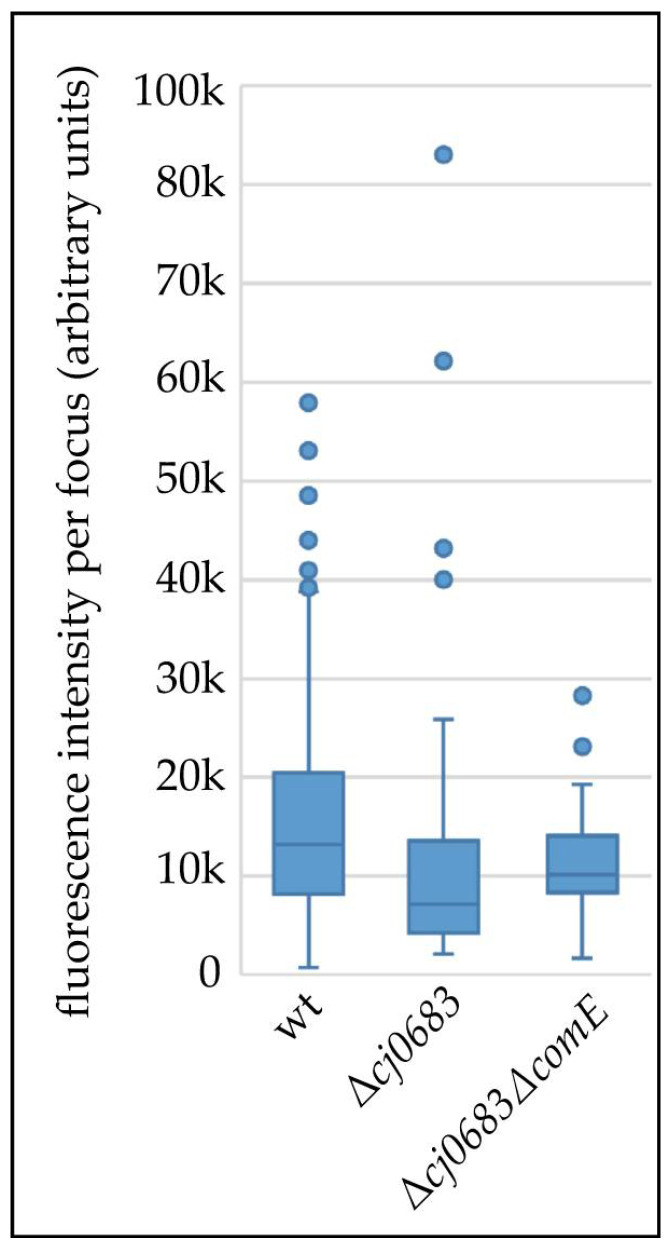
The median amount of DNA within single foci is only slightly reduced in rare uptake events of Δ*cj0683* and Δ*cj0683*Δ*comE* compared to the wild type. The boxplot length corresponds to the interquartile range (IQR; 50%) of data, the horizontal bar indicates the median value; whiskers represent 1.5 × IQR or the maximum/minimum value of the dataset; dots, outliers. Δ*cj0683* (*n* = 65 foci); Δ*cj0683*Δ*comE* (*n* = 37 foci); wild type (*n*= 739 foci).

## Data Availability

Data is provided in the manuscript. Further data that support the findings of this study is available on request from the corresponding author.

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
