# Peer review of "Cj0683 Is a Competence Protein Essential for Efficient Initialization of DNA Uptake in Campylobacter jejuni"

_biomolecules, 2023, doi:10.3390/biom13030514_

Round 1

Reviewer 1 Report

The article “DNA uptake capacity of Campylobacter jejuni is dependent on the periplasmic protein Cj0683 rather than ComE” by Julia C. Golz, Sandra Preus, Christoph Puning, Greta Golz and Kerstin Stingl, aims to provide some details to the known mechanism of DNA uptake and transformation in C. jejuni, an important food-borne pathogen displaying high genetic diversity caused, among others, by a phenomenon of natural transformation. The authors use a previously validated method of DNA uptake observation at the single cell level in C. jejuni and estimate the DNA transformation capacity of C. jejuni to be around 30 kb per uptake location. The authors also recapitulate the mutagenesis of genes encoding components of the uptake and transformation system and show the negligible role of one of its components, ComE protein, in the process in C. jejuni. A role in periplasmic transport of DNA was proposed for novel periplasmic protein Cj0683.

General comments

The manuscript is relevant for the field and presented in a quite well-structured manner. Experiments have appropriate controls and are presented in clear-form figures. However, extensive native English speaker editing should be considered. The main weakness of the presented manuscript is that a huge work was done on the mutagenesis and testing of the phenotypic consequences of deletions of pilQ, and comEC, while the authors themselves underline in the introduction section that such mutants were previously constructed, and specific roles of encoded proteins in the DNA transport into the periplasm and further into the cytoplasm were shown. Thus, these genes should not be called “putative players” (or “genes with potential roles”) if their specific function was shown and appropriate references cited. Testing the previously implemented method of monitoring of fluorescent DNA uptake could be a minor part of this work, while showing that the role of ComE is negligible in periplasmic DNA transport and Cj0683 protein plays this role instead, should be highlighted more - it brings novel information about the transformation in this important group of pathogenic bacteria. The discussion section should be more focused on discussing specific results with available data and not recapitulating the results; unfortunately, this section needs some improvements

1. Is the method used for generation of deletion mutants a routine method in this species?

2. Why do authors complement mutation by inserting a gene in a neutral location where transcription is directed by a cassette promoter, and then try to delete the gene from the native location? This procedure would be justified if the were indications that mutations are lethal. Is there any other justification for not using a vector-based method for introduction of the missing gene in trans to the cells of the previously created mutants? If the method is specific for this bacterium/ group of bacteria, the authors are just asked to specify this in the methods section.

3. Why the authors check if competence supports biofilm formation? The extracellular DNA is present within the biofilm, so it may be the source of DNA taken up by the cells residing in the biofilm, but why would biofilm formation be dependent on active DNA transport and transformation?

Specific comments

1. Introduction does not give sufficient background in the field; moreover, the authors are encouraged to add a scheme showing the elements of the DNA uptake and transport in Gram-negative vs. Gram-positive bacteria (the latter are also mentioned in the article); the literature is not most recent – only 11 out of 27 references were published within the last 10 years, and 7/21 within last 5 years.

2. The aim of the study should underline the novelty of the work with quantitative comparisons with H. pylori and functional analyses of the novel component of the system, i.e. Cj0683 protein.

3. Materials and methods section is detailed enough to repeat experiments. In fact, the section could be shorter and more concise, while still containing enough details to repeat experiments. The method of gDNA isolation should be indicated. Is the genomic DNA used for transformation somehow fragmented – most isolation method result in DNA which is fragmented due to physical forces during processing; the authors mention that they estimate the length of fragments for C. jejuni gDNA ~20kb; is it normal after purification with a particular method used?

4. All the strains used in this work should be described in the first section of the M&M, e.g. NCTC 11168 name appears somewhere further in the text without any prior description.

5. The description of construction of deletion mutants is difficult to follow; why was kanamycin cassette used form some genes and gentamicin for others? Why only cj0683 fusion fragment was methylated? Results of PCR checking for the results of allelic exchange should be given in supplementary file.

6. L234-236: what is meant by “how active distinct DNA…”?

7. L234-236: “…if multiple DNA molecules were imported.” – does it mean at one location?

8. Figure 1: C should be indicated in the figure.

9. L339: there is homology or there is not; acceptable could be the term “distant homology”, but its measure is identity/similarity between amino acid sequences of compared proteins.

10. Figure 4 and appropriate part of the text: if pilQ determines import of DNA into the periplasm, why is DNA uptake level in the complemented mutant close to none? If complementation was successful, the phenotype, in this case transport of DNA to the periplasm, should be significantly restored. Further, DNA uptake was not restored while transformation rate was quite high? Why would a reintegration of pilQ occur from hsdM into native location – and even if it is possible, why would that affect the phenotype of the strain – there still would be present a copy of pilQ? Maybe introduction of this gene on a plasmid would be a solution? Or maybe the protein is not properly targeted to the outer membrane and does not form a functional pore?

11. L368-373: the results presented in this paragraph should be shown; otherwise, the results of complementation trial does not prove the specific role of pilQ

12. The links and references for the bioinformatic tools should be given in materials and methods, and removed from this section (L396, 403-404, 407)

13. L398: This paragraph should begin with a short information how the authors come on this Cj0683 protein as a possible player in transformation in C. jejuni?

14. L417, 419: “complementation mutant” is not correct, the “complemented mutant” or “complemented strain” is correct

15. Comparing the results for comE mutant in Figure 4 and cj0683 mutant in Figure 6: is it possible that these proteins interact and act together at this stage of DNA uptake?

16. 4.3. part of discussion section should be the most emphasized as it gives the most novel information

17. L524-526: Maybe some suppression mutation has occurred? Rareness suggest such probability.

18. L531-534: Are Cj0683 and ComH similar (amino acid sequence, predicted structure etc.) to support such speculations?

Author Response

We thank the reviewer for the valuable comments which we answer point-by-point in the following in bold italics. Line numbers refer to the pdf document with track changes.

Reviewer #1

The article “DNA uptake capacity of Campylobacter jejuni is dependent on the periplasmic protein Cj0683 rather than ComE” by Julia C. Golz, Sandra Preus, Christoph Puning, Greta Golz and Kerstin Stingl, aims to provide some details to the known mechanism of DNA uptake and transformation in C. jejuni, an important food-borne pathogen displaying high genetic diversity caused, among others, by a phenomenon of natural transformation. The authors use a previously validated method of DNA uptake observation at the single cell level in C. jejuni and estimate the DNA transformation capacity of C. jejuni to be around 30 kb per uptake location. The authors also recapitulate the mutagenesis of genes encoding components of the uptake and transformation system and show the negligible role of one of its components, ComE protein, in the process in C. jejuni. A role in periplasmic transport of DNA was proposed for novel periplasmic protein Cj0683.

General comments

The manuscript is relevant for the field and presented in a quite well-structured manner. Experiments have appropriate controls and are presented in clear-form figures. However, extensive native English speaker editing should be considered. The main weakness of the presented manuscript is that a huge work was done on the mutagenesis and testing of the phenotypic consequences of deletions of pilQ, and comEC, while the authors themselves underline in the introduction section that such mutants were previously constructed, and specific roles of encoded proteins in the DNA transport into the periplasm and further into the cytoplasm were shown. Thus, these genes should not be called “putative players” (or “genes with potential roles”) if their specific function was shown and appropriate references cited. Testing the previously implemented method of monitoring of fluorescent DNA uptake could be a minor part of this work, while showing that the role of ComE is negligible in periplasmic DNA transport and Cj0683 protein plays this role instead, should be highlighted more - it brings novel information about the transformation in this important group of pathogenic bacteria. The discussion section should be more focused on discussing specific results with available data and not recapitulating the results; unfortunately, this section needs some improvements

>We are grateful for the reviewer’s suggestions. The function of a knock-out mutant of pilQ and comEC homologs in C. jejuni were previously analyzed using the classical approach of counting transformants after selection of a resistance marker. With our single cell assay we add information about the direct role in either outer membrane transport or downstream processes. We agree that the novelty of this study is predominantly the quantification of DNA uptake in C. jejuni and the discovery of the periplasmic protein Dcj0683 with putative function as pilus protein for DNA binding and transfer to the outer membrane, while demonstrating the negligible role of ComE in DNA uptake in C. jejuni. As requested by the reviewer, we changed the title, reformulated the abstract and the last chapter in the introduction, added more data on a Dcj0683DcomE double mutant (in Fig. 6) and added a new Figure 7 with quantitative data on the residual rare events observed in Dcj0683 and Dcj0683DcomE. The combined results suggest that those rare events led to similar DNA uptake amounts compared to the wild type. Further data are provided in order to exclude suppression mutations to have occurred in those bacteria displaying rare events (see below). Thus, we concluded that Cj0683 increased the probability of initiation of DNA uptake events, which would be consistent with a role as putative DNA binding pilus protein.

  1. Is the method used for generation of deletion mutants a routine method in this species?

> Deletion of the target gene by insertion of a resistance cassette upon homologous recombination is a routine method in C. jejuni. However, the introduction of the construct can be introduced into C. jejuni by two different ways, either electroporation of non-methylated suicide plasmids or natural transformation of methylated DNA. The study was started before the publication of Beauchamp et al. 2017, in which methylation was identified essential for DNA uptake via natural transformation. Before, electroporation was the method of choice for construction of C. jejuni mutants, which is, however, far less efficient than natural transformation. Thus, the three deletion mutants were constructed by electroporation. Later on, the complementation mutants and the novel cj0863 mutant was constructed by natural transformation. We added an explanatory sentence to the Material & Methods part (line 198-209).

  1. Why do authors complement mutation by inserting a gene in a neutral location where transcription is directed by a cassette promoter, and then try to delete the gene from the native location? This procedure would be justified if the were indications that mutations are lethal. Is there any other justification for not using a vector-based method for introduction of the missing gene in trans to the cells of the previously created mutants? If the method is specific for this bacterium/ group of bacteria, the authors are just asked to specify this in the methods section.

> There are two reasons to avoid a vector-based method. First, C. jejuni 81-176 already carries plasmids, thus, introduction of another plasmid is complicated and usually not done. Second, expression from multi-copy plasmids is even worse for membrane proteins, which are expected to be non-functional if overexpressed (non-functional inclusion bodies will be produced). The reviewer is right that introduction of a second copy of a gene prior to its deletion at the native location is commonly done if the target gene is essential for growth BUT also if the target gene is essential for transformation. Otherwise reintroduction of the target gene can only performed by electroporation but not via the more efficient way of natural transformation (see also below). That is why we first introduced a second copy of pilQ and comEC prior to deletion of the target genes at the native locus. We added an explanation in lines 232-234.  

  1. Why the authors check if competence supports biofilm formation? The extracellular DNA is present within the biofilm, so it may be the source of DNA taken up by the cells residing in the biofilm, but why would biofilm formation be dependent on active DNA transport and transformation?

> Capacity for natural transformation and biofilm formation was previously suggested to be interconnected [added references 27-29]. Also external DNA serves as extracellular matrix supporting biofilm formation. Thus, we wondered whether binding and/or uptake of DNA does support biofilm formation in C. jejuni. We added more background information in the introduction lines 78-83 and also discussed the issue in the discussion chapter 4.3 (lines 620-632).

Specific comments

  1. Introduction does not give sufficient background in the field; moreover, the authors are encouraged to add a scheme showing the elements of the DNA uptake and transport in Gram-negative vs. Gram-positive bacteria (the latter are also mentioned in the article); the literature is not most recent – only 11 out of 27 references were published within the last 10 years, and 7/21 within last 5 years.

>As requested we added additional background in the introduction (references 3,7,10,15-17,25,27-29). However, since the manuscript rather focusses on novel aspects of DNA import in C. jejuni and should not be considered a review article, we think that a schematic overview of DNA import machineries of Gram-negative and –positive bacteria is not suitable. We instead added references of appropriate recent review articles (references 9,10) and added more in-text information about the DNA mechanism over the outer and inner membrane. We hope to present a solid background of information helpful to place our new results into context. 

  1. The aim of the study should underline the novelty of the work with quantitative comparisons with H. pylori and functional analyses of the novel component of the system, i.e. Cj0683 protein.

> As mentioned above, we changed the title, reformulated the abstract and the last chapter in the introduction (lines 84-94), add data on a Dcj0683DcomE double mutant (in Fig. 6) and a new Figure 7 with quantitative data on the residual rare events in Dcj0683 and Dcj0683DcomE. We extended our conclusion that Cj0683 increased the likelihood of initiation of DNA uptake events, which would be consistent with a role in DNA acquisition putatively in association with a postulated (pseudo-)pilus.

  1. Materials and methods section is detailed enough to repeat experiments. In fact, the section could be shorter and more concise, while still containing enough details to repeat experiments. The method of gDNA isolation should be indicated. Is the genomic DNA used for transformation somehow fragmented – most isolation method result in DNA which is fragmented due to physical forces during processing; the authors mention that they estimate the length of fragments for C. jejuni gDNA ~20kb; is it normal after purification with a particular method used?

>We shortened this section by moving the oligo list to the supplementary material and by skipping details on electroporation (referring to [36]) and reformulating sentences on the construction of strains (a schematic overview is now also presented in Figure S1). The DNA isolation method was added in lines 122-123. According to manufactures data sheet the isolated DNA is 20-50 kb, which is common for silica column based methods. 

  1. All the strains used in this work should be described in the first section of the M&M, e.g. NCTC 11168 name appears somewhere further in the text without any prior description.

> We inserted Table S2 with all strains used in this study in the Supplementary Material

The description of construction of deletion mutants is difficult to follow; why was kanamycin cassette used form some genes and gentamicin for others? Why only cj0683 fusion fragment was methylated? Results of PCR checking for the results of allelic exchange should be given in supplementary file.

> The chapter 2.3 of strain construction was reformulated. The apramycin resistance was used for Dcj0683 in order to be able to also create the double mutant DcomEDcj0683. Those data are included now in Figure 6 and in the new Figure 7. Results of PCR checking for the successful allelic exchange are given in Figure S1.

  1. L234-236: what is meant by “how active distinct DNA…”?

> We changed the sentence to “how much DNA was imported at distinct DNA uptake locations”.

  1. L234-236: “…if multiple DNA molecules were imported.” – does it mean at one location?

> Yes, we reformulated “…if multiple DNA molecules were imported at one location”.

  1. Figure 1: C should be indicated in the figure.

> “C” was added to Figure 1.

  1. L339: there is homology or there is not; acceptable could be the term “distant homology”, but its measure is identity/similarity between amino acid sequences of compared proteins.

> We are grateful for this comment. We changed the term “weak homology” to “distant homology”.

  1. Figure 4 and appropriate part of the text: if pilQ determines import of DNA into the periplasm, why is DNA uptake level in the complemented mutant close to none? If complementation was successful, the phenotype, in this case transport of DNA to the periplasm, should be significantly restored. Further, DNA uptake was not restored while transformation rate was quite high? Why would a reintegration of pilQ occur from hsdM into native location – and even if it is possible, why would that affect the phenotype of the strain – there still would be present a copy of pilQ? Maybe introduction of this gene on a plasmid would be a solution? Or maybe the protein is not properly targeted to the outer membrane and does not form a functional pore?

> Complementation of membrane proteins is tricky, since they often do not act alone but are integrated in protein complexes with a distinct stoichiometry of subunits. Complementation of pilQ from a plasmid would lead to even higher levels of PilQ production. In the discussion chapter we explain this issue and refer to partially complemented strains of other groups working on natural transformation of C. jejuni (Wiesner et al. 2003) in lines 602-606. It should also be considered that the microscopically observed fraction of cells with active DNA uptake is measured on a linear scale (in %). Transformation rates are measured in log-scale. Therefore, we argue in our discussion that the ΔpilQ-compl strain was similarly restored when comparing the fraction of competent cells and the transformation rate (lines 600-602). Furthermore, DNA amounts per focus imported in the ΔpilQ-compl strain were comparable to the wild type (Figure S2, lines 441-443).

  1. L368-373: the results presented in this paragraph should be shown; otherwise, the results of complementation trial does not prove the specific role of pilQ

> The data of DNA uptake from two successfully transformed 81-176ΔpilQ-compl-strep mutants is added in Figure S3.

  1. The links and references for the bioinformatic tools should be given in materials and methods, and removed from this section (L396, 403-404, 407)

> We moved this information into the material and methods section 2.5 (line 292-299).

  1. L398: This paragraph should begin with a short information how the authors come on this Cj0683 protein as a possible player in transformation in C. jejuni?

> We provide the respective information in lines 484-501.

  1. L417, 419: “complementation mutant” is not correct, the “complemented mutant” or “complemented strain” is correct

> Corrected as requested by the reviewer.

  1. Comparing the results for comE mutant in Figure 4 and cj0683 mutant in Figure 6: is it possible that these proteins interact and act together at this stage of DNA uptake?

> We did not analyse interaction of ComE and Cj0683. But we guess the reviewer asks whether ComE can partially take over the function of Cj0683 and whether the rare DNA uptake events resulting in very low transformation rate observed for the deletion mutant of Dcj0683 is dependent on ComE presence. We actually had the same question and constructed a double mutant lacking ComE and Cj0683, for which we added the data in Figure 6 and Figure 7. This double mutant showed a comparable fraction of bacteria with DNA uptake (rare events) when compared to the single Dcj0683 mutant, showing that ComE is not involved in residual DNA uptake events in the Dcj0683 mutant. However, as already observed for the DcomE mutant, which had slightly decreased transformation rate compared to the wild type, transformation rate in the double mutant was 5-fold reduced compared to the single Dcj0683 mutant (Fig. 6B), hinting at a downstream role of ComE like e. g. protection of incoming DNA and/or transfer to the ComEC channel.

  1. 4.3. part of discussion section should be the most emphasized as it gives the most novel information

> We are grateful for this comment. As mentioned above we emphasized the novelties of our study in the altered title, the abstract and the introduction part, optimized the material and method part, integrated new data in the results part on the double mutant of Dcj0683DcomE in Figure 6 and present a new Figure 7 with quantitative information on the rare events found in the Dcj0683 mutant and the double mutant. We think that we thereby improved the understanding of our study.

  1. L524-526: Maybe some suppression mutation has occurred? Rareness suggest such probability.

> To exclude suppression mutation in those Dcj0683bacteria displaying functional DNA uptake (rare events), we transformed the Dcj0683 with the streptomycin resistance marker and selected the rare Strep-resistant transformants. If suppression mutations had occurred in these bacteria, we expected a wild type phenotype of DNA uptake. For this purpose, the Dcj0683-strep clones were analyzed for DNA uptake and we indeed identified the same rare events as observed in the parental Dcj0683 strain (Figure S3). In addition, we analysed the amount of DNA imported during these rare events and observed similar uptake capacity per DNA focus (Figure 7). Thus, we conclude that the phenotype of Dcj0683 and its derivative clones Dcj0683-strep was due to absence of Cj0683 and that absence of Cj0683 led to drastically decreased DNA uptake probability, consistent with a putative role of Cj0683 as part of the (pseudo-)pilus involved in acquisition and initiation of DNA uptake but not in force-generation as proposed for ComE in other bacteria. We updated chapter 3.4 and 4.4.

  1. L531-534: Are Cj0683 and ComH similar (amino acid sequence, predicted structure etc.) to support such speculations?

> No, as mentioned in line 494-495 the Cj0683 protein is unique for Campylobacter spp. Actually, this is also the case for ComH, which is unique for Helicobacter pylori (lines 63-65). We tried to emphasize that Cj0683 probably plays a different role than ComE (force generation for DNA uptake into the periplams) or ComH (transfer of already imported DNA to the ComEC channel) (lines 672-675). We alternatively propose a role in DNA acquisition to the cell surface and/or initialization of DNA import (consistent with Cj0683 beeing part of the putative (pseudo-)pilus), since rare events of DNA uptake in the absence of Cj0683 were not distinguishable from the wild type events.  

Reviewer 2 Report

In this study, the authors characterized the mechanism underlying natural transformation in Campylobacter jejuni.

Major results are:

(1) The authors developed a method to measure the DNA uptake from outside of the cells to periplasm using fluorescent DNAs. Using this method, the mechanism of DNA uptake was characterized.

(2) Among homolog genes involved in type II secretion /typeIV pilus system which are important for natural competence, comE only limitedly contributed whereas pilQ and comEC are essential.

(3) The DNA uptake did not affect biofilm formation in C. jejuni.

(4) A novel periplasmic protein Cj0683, instead of comE, was revealed to be important for natural competence.

Experiments are rationally designed.

The manuscript is well written and reader-friendly.

Some concerns, nevertheless, still remain:

Line 350-351

Figure 4

Was the appearance of transformants in delta-pilQ and delta-comEC below the detection limit?  

Alternatively, were some transformants detected?

Line 374-390

There is no description of the impact of DNA uptake on biofilms in Discussion.

The authors should refer to previous papers on Campylobacter biofilm formation and discuss the biological and pathological implications of the DNA uptake.

Additionally, related references and information should be provided in Introduction.

Figure 1

Panel C is not indicated.

Line 394-396

Defining Cj0011c as comE may be inconsistent with the statement in this section.

Figure 5

Statistical analysis is required.

If available, experimental results should be evaluated using mutants that cannot form biofilms (e.g., flaA- and motA-) as controls.

Figure 6

Discuss why the transformation rate is still higher in delta-cj0683 than in delta-pilQ.

//

Author Response

We thank the reviewer for the valuable comments which we answer point-by-point in the following in bold italics. Line numbers refer to the pdf document with track changes.

Reviewer #2

In this study, the authors characterized the mechanism underlying natural transformation in Campylobacter jejuni.

Major results are:

(1) The authors developed a method to measure the DNA uptake from outside of the cells to periplasm using fluorescent DNAs. Using this method, the mechanism of DNA uptake was characterized.

(2) Among homolog genes involved in type II secretion /typeIV pilus system which are important for natural competence, comE only limitedly contributed whereas pilQ and comEC are essential.

(3) The DNA uptake did not affect biofilm formation in C. jejuni.

(4) A novel periplasmic protein Cj0683, instead of comE, was revealed to be important for natural competence.

Experiments are rationally designed. The manuscript is well written and reader-friendly.Some concerns, nevertheless, still remain:

Line 350-351, Figure 4

Was the appearance of transformants in delta-pilQ and delta-comEC below the detection limit?  Alternatively, were some transformants detected?

> We did not observe any transformants, when we added streptomycin resistance marker to the delta-pilQ or the delta-comEC mutant. Our detection limit was 8.5 x 10-9 ± 2.4 x 10-9 (now added in lines 420-421). Thus, both mutants were completely deficient of DNA uptake and of natural transformation.

Line 374-390

There is no description of the impact of DNA uptake on biofilms in Discussion. The authors should refer to previous papers on Campylobacter biofilm formation and discuss the biological and pathological implications of the DNA uptake. Additionally, related references and information should be provided in Introduction.

> We added information of DNA uptake on biofilms in the introduction (lines 78-83) and discussed our results in the context of previous papers in lines 620-632.

Figure 1, Panel C is not indicated.

> Panel C was added in Figure 1. 

Line 394-396

Defining Cj0011c as comE may be inconsistent with the statement in this section.

> Cj0011 was defined as ComE according to homology between the proteins (lines 69-70, 413-415). The function of ComE in C. jejuni is different from other Gram-negative bacteria. It is not the force-generation protein for import of DNA into the periplasm. However, we observed a small effect on transformation rate (50-fold reduction in the deletion mutant compared to the wild type and 5-fold reduction in the double Dcj0683DcomE mutant compared to the single Dcj0683 mutant), which might be due to a role of ComE in DNA protection and/or transfer to the inner membrane channel (now added in lines 534-537, 617-619). 

Figure 5, Statistical analysis is required. If available, experimental results should be evaluated using mutants that cannot form biofilms (e.g., flaA- and motA-) as controls.

 >Statistical analysis was added to Figure 5. Unfortunately, we have not further data on flagellar mutants as negative controls. But we discuss our results in the context of previous papers, showing similar biofilm capacity using the same assay (reference 38).

Figure 6

Discuss why the transformation rate is still higher in delta-cj0683 than in delta-pilQ.

> PilQ is essential for DNA uptake. Lack of Cj0683 drastically reduced DNA uptake probability, while rare events were still observed (Figure 6), which displayed similar amounts of DNA imported per focus (new Figure 7). We conclude that absence of Cj0683 led to drastically decreased DNA uptake probability, consistent with a putative role of Cj0683 as part of the (pseudo-)pilus involved in acquisition and initiation of DNA uptake but not in force-generation as proposed for ComE in other bacteria.

Round 2

Reviewer 1 Report

The authors should be commended not only for their compliance or substantive discussion with the reviewers' comments, but also for their own initiative, which significantly contributed to the quality and readability of the article. I have no additional comments.